# Consolidation in a crisis: Patterns of international collaboration in early COVID-19 research

**Caroline V. Fry[1¤], Xiaojing Cai[2,3], Yi Zhang[4], Caroline S. Wagner[2]\***

**1** Massachusetts Institute of Technology, Cambridge, MA, United States of America, **2** John Glenn College of Public Affairs, The Ohio State University, Columbus, OH, United States of America, **3** School of Public Affairs, Zhejiang University, Hangzhou, Zhejiang, China, **4** Centre for Artificial Intelligence, University of Technology Sydney, Ultimo, Australia

¤ Current address: Department of Management and Industrial Relations, Shidler College of Business, University of Hawai'i at Manoa, Honolulu, Hawaii
\* wagner.911@osu.edu

**Data Availability Statement:** We have placed a cleaned data set on figshare. https://figshare.com/articles/COVID19_cooperative_research_project/12221093.

## Abstract

This paper seeks to understand whether a catastrophic and urgent event, such as the first months of the COVID-19 pandemic, accelerates or reverses trends in international collaboration, especially in and between China and the United States. A review of research articles produced in the first months of the COVID-19 pandemic shows that COVID-19 research had smaller teams and involved fewer nations than pre-COVID-19 coronavirus research. The United States and China were, and continue to be in the pandemic era, at the center of the global network in coronavirus related research, while developing countries are relatively absent from early research activities in the COVID-19 period. Not only are China and the United States at the center of the global network of coronavirus research, but they strengthen their bilateral research relationship during COVID-19, producing more than 4.9% of all global articles together, in contrast to 3.6% before the pandemic. In addition, in the COVID-19 period, joined by the United Kingdom, China and the United States continued their roles as the largest contributors to, and home to the main funders of, coronavirus related research. These findings suggest that the global COVID-19 pandemic shifted the geographic loci of coronavirus research, as well as the structure of scientific teams, narrowing team membership and favoring elite structures. These findings raise further questions over the decisions that scientists face in the formation of teams to maximize a speed, skill trade-off. Policy implications are discussed.

## Introduction

The global pandemic caused by the widespread coronavirus, COVID-19, stimulated extraordinary amounts of scientific inquiry around the world. The virus first appeared in the scholarly literature on the 24 January 2020 [1], and subsequently, virologists and immunologists worked to isolate and identify the virus, determine its etiology, define the vulnerabilities that may allow treatment, and conduct research on drug and vaccine development. While international

**Funding:** This study was supported by the Fulbright Association and the Natural Science Foundation of China (71843012). The funders had no role in study design, data collection and analysis, decision to publish, or preparation of the manuscript.

**Competing interests:** The authors have declared that no competing interests exist.

collaboration and cooperation are critical actions to address global pandemics, the need for rapid and urgent solutions could render cross-border teamwork more difficult, due to the transaction costs of communication and rising political tensions. This article tracks patterns of international collaboration in coronavirus-related research before and in the period immediately after the start of the COVID-19 pandemic in order to understand how scientists leveraged complementary expertise within their own nation's borders and beyond.

International collaboration in scientific research has grown at a spectacular level since the 1980s, when geopolitical shifts opened up opportunities for formerly restricted researchers to create relationships outside their nation or region [2,3]. The breakup of the Soviet Union, the reunification of Germany, and China's decision to create the "four modernizations" to include science and technology in a process of opening up (改革开放), all served to restructure science. In particular, collaborations between China and the United States have grown rapidly, and the rate of collaboration between these two countries are now more numerous than any two countries in the world [2].

While international collaborations can help scientists in one country to access complementary expertise outside their country's borders, there are search and coordination costs associated with such collaborations [4–6]. International collaborative research activities operate as a network [7] which takes time to traverse. No international organization oversees or directs these works: researchers find each other based on shared interests and the needs of frontier science. This is particularly true for sciences of immunology and virology, where no central laboratory or common data set is on hand as an organizing force [7].

COVID-19, the coronavirus that emerged in late 2019 and grew to a global pandemic in early 2020, presented this trade-off between novelty and efficiency to the international community of scholars. The importance of global collaboration and coordination to resolve this enormous challenge is summarized by the director of the United States National Institutes of Health (NIH), stating:

*"We need to bring the full power of the biomedical research enterprise to bear on this crisis. Now is the time to come together with unassailable objectivity to swiftly advance the development of the most promising vaccine and therapeutic candidates that can help end the COVID-19 global pandemic."*

NIH Director Dr. Francis S. Collins. April 2020

But at the same time, the urgent need for solutions to combat the pandemic increases the cost of search and coordination needed in internationally collaborative work.

In a time of urgency, we expect that scientists reduce their collaborations, or seek to work with known colleagues to reduce the transaction costs of communication. We hypothesize that the pressures presented by the coronavirus crisis would lead scientists to collaborate internationally at a lower rate than before the pandemic. We expect search to be reduced, and pre-existing relationships to be strengthened, to the exclusion of scientists from less developed institutions, regions, or nations. In addition to a test of these hypothesis, in this study we explore whether these shifts during the early months of the pandemic altered the geographic locus of coronavirus research, and whether there are implications for the quality, and type, of work produced.

## Methods

In order to achieve the goals of the study, the project team constructed two brand new datasets: one to capture measures on collaborations in coronavirus research prior to the COVID-19 crisis, and one for the COVID-19 crisis period. The pre-COVID-19 period extends for 24 months

prior to December 2019. The COVID-19 period extends from January 1, 2020 to April 23, 2020. Measures of collaboration (and others) are generated using scientific articles.

A complete dataset of scientific articles on coronavirus-related research between January 1, 2018 and April 23, 2020 are extracted from the Clarivate Web of Science (WoS), Elsevier Scopus, and PMC-sourced materials drawn from CORD-19 (COVID-19 Open Research Dataset). Any overlap between articles found across the different source materials were removed. To complement the data on published articles, we also drew preprint articles between January 1 and April 23 2020 from bioRxiv.org, medRxiv.org, and arXiv.org, extracted through the Dimensions database. The following consistent set of keywords was used in searches in the Title/Abstract/Keywords of each article in the respective databases:

- "COVID-19" OR "2019-nCoV" OR "coronavirus" OR "Corona virus" OR "SARS-CoV" OR "MERS-CoV" OR "Severe Acute Respiratory Syndrome" OR "Middle East Respiratory Syndrome"

Table 1 shows the composition of the datasets in the pre-COVID-19 and during COVID-19. Using the search procedure outlined, the data comprise a total of 10,432 coronavirus-related articles and preprints with author identifiable information across the two periods for analysis. Within these data, 5,934 articles are published in peer-reviewed journals; the remainder of the works comprise reviews, conference proceedings, and preprints (considered 'informal' hereafter).

Pre-COVID-19 analysis is limited to published records because the historical data are standardized in indexed databases. This allows others to recreate the dataset and test the validity of this analysis. Moreover, enough time had passed to allow most works from pre-COVID-19 to be peer reviewed and published in recognized scholarly venues (an anonymized dataset for this project are also made available on figshare). The COVID-19 period includes both peer-reviewed and preprint materials. We include preprint materials because the time pressures imposed by the pandemic crisis propelled ready and open sharing of even initial results: researchers put materials into circulation to provide insights for others without waiting for peer review. This process means that, at the time of this writing, many coronavirus-related articles have not had time to be peer reviewed and published in established venues. By necessity, this means that the COVID-19 dataset includes materials that are questionable in their scientific rigor and that may be methodologically unsound. Future work will return to the materials to re-evaluate the published record for those materials that have failed to make it through the peer-review process.

The data are examined for several features: 1) publication patterns and numbers; 2) public funding patterns to compare pre- and COVID-19, as available; 3) the structure of scientific teams; 4) quality measures of formal publications; 5) collaborative patterns at the international level; and 6) networked collaborations including measures of egonets at the international level.

**Table 1. Data source and publication data.**

| Source | Pre-COVID-19 (January 1st 2018 –December 31st 2019) | COVID-19 |
|---|---|---|
| Scopus | 1,917 | 1,714 |
| Web of Science | 1,448 | 822 |
| PMC | 4,198 | 4,334 |
| Preprints (BioRxiv/MedRxiv) | | 2,147 |
| Combined (duplicates dropped) | 6,337 | 5,083 |
| Combined, with author affiliation data | 6,105 | 4,327 |

For each article in the datasets, variables of interest are created based on author institutional affiliation, publication journal impact measures, and funding support.

To test any changes in publication patterns and team structure between articles in the pre- and COVID-19 period, we run a series of tests to ascertain any statistically significant changes. Specifically, we use a combination of one-tailed T-tests and ordinary least squares to ascertain any average differences in article features between the two groups. The one-tailed T-tests compare the mean of variables of the two groups and using the standard deviation of the two samples allows us to assess whether the variables in question come from the same distribution, or different. Statistical significance is assessed at the 0.1, 0.05, and 0.01 level.

To test any dynamic change in network structure of researchers between pre-COVID-19 and during COVID-19, we construct global collaboration networks based on international coauthorships. Collaboration links are first established by addresses in articles, based on a full counting method. Then, co-occurrence matrices are created to show which countries are coauthoring articles together. The coauthorship links are aggregated by country pairs and imported into software VOSviewer [8] and Gephi [9] for network analysis that allows for a statistical review of the whole sample of collaborating countries and allows visualization of the connections. To assess any changes in network positions of nations, we calculate several network metrics for selected nations in both periods, namely: a. Degree, the number of connecting nodes or collaboration partners of a focal country; b. Weighted Degree, a measure of the number of collaboration links a nation has [10]; c. Normalized Betweenness Centrality, a measure of how often a node appears on the shortest path between other nodes in the network [11] and Eigenvector Centrality measures the influence of hubs in a network [12].

Finally, in order to visualize the landscape of the COVID-19 research and compare any change in research interest between the pre- and COVID-19 period from the perspective of topic analysis, we conduct a keyword-based bibliometric analysis to generate a co-term network for the two periods respectively. With the aid of VantagePoint—a software platform for bibliometrics-based text analytics owned by Search Technology Inc, we collect COVID-19-related core terms by exploiting a term clumping process [13] and create co-term networks, in which each node represents a core term and each edge reflects the co-occurrent frequency of its connected terms. VOSviewer [8] is used for visualizing the networks in the form of science maps.

## Results

The aforementioned propositions are tested using statistical and network tests designed to ascertain differences in publication patterns, the structures of teams and international rates of collaboration in coronavirus-related research before and during the COVID-19 global pandemic.

### National contributions to coronavirus articles

Following the COVID-19 outbreak, as expected given the geographic spread of the pandemic [14], coronavirus-related articles are more likely to be authored by scientists based in China and Italy than before the outbreak. As the two nations that experienced the earliest outbreaks of the virus, this suggests that a need for solutions and access to patient populations can stimulate research productivity in a topic. In contrast, fewer articles emanate from other OECD or developing countries in the early COVID-19 period as compared to before the pandemic. Table 2 compares the geographic sources of coronavirus research in the pre- and COVID-19 research datasets. It is clear that China takes the lead on research publications during the COVID-19 period, with the percentage of Chinese articles growing to 39% from 22% prior to

**Table 2. Author location in COVID-19 vs pre-COVID-19 research.**

| | Pre-COVID-19 (N = 6,105) | | | COVID-19 (N = 4,327) | | | | COVID-19 (minus preprints) (N = 2,472) | | | |
|---|---|---|---|---|---|---|---|---|---|---|---|
| | Mean | Median | Std Dev | Mean | Median | Std Dev | p-value | Mean | Median | Std Dev | p-value |
| **Authors China** | 0.22 | 0 | 0.41 | 0.39*** | 0 | 0.49 | .000 | 0.42*** | 0 | 0.49 | .000 |
| **Authors OECD** | 0.67*** | 1 | 0.47 | 0.58 | 1 | 0.50 | .000 | 0.54 | 0 | 0.50 | .000 |
| **Authors United States** | 0.35*** | 0 | 0.48 | 0.28 | 0 | 0.45 | .000 | 0.25 | 0 | 0.43 | .000 |
| **Authors United Kingdom** | 0.090 | 0 | 0.29 | 0.098 | 0 | 0.30 | .091 | 0.096 | 0 | 0.29 | .207 |
| **Authors Italy** | 0.035 | 0 | 0.19 | 0.051*** | 0 | 0.22 | .000 | 0.059*** | 0 | 0.24 | .000 |
| **Authors Asia, not China** | 0.27*** | 0 | 0.44 | 0.17 | 0 | 0.37 | .000 | 0.19 | 0 | 0.39 | .000 |
| **Authors, not China NOR Europe NOR OECD** | 0.15*** | 0 | 0.35 | 0.11 | 0 | 0.32 | .000 | 0.12 | 0 | 0.32 | .000 |

*, **, *** denote statistical significance at p-values of 0.1, 0.05 and 0.01 in a difference of means test comparing pre- and COVID-19 outcomes. Comparisons are between pre-COVID-19 outcomes and COVID-19 era outcomes.

the outbreak, while the United States' output drops as a share of total output during the COVID-19 period.

Table 3 provides a comparison of the quantity of articles produced by China and the United States in the pre- and COVID-19 periods available at the time of writing. The number of articles produced by Chinese authors in the first three months of the COVID-19 period—more than 1,600 articles—surpasses the number of coronavirus articles produced by Chinese-based authors in the entire previous 24 months. As a preliminary exploration into the extent of China's involvement in international collaborative research Table 3 also reveals that by April 2020, Chinese authors together with international collaborators produced over 12% of articles in the topic of 'coronavirus'–again, more than the volume that they produced across 2018 and 2019 together. In contrast, United States-based scholars produced just under half of the volume combined of international collaborative research that they produced in 2018 and 2019. This finding is explored further in subsequent sections of the paper.

We turn our analysis from the national to the institutional level in Fig 1, and assess which institutions are the top producers of coronavirus research in the pre- and COVID-19 period. Consistent with Tables 2 and 3, we find that Chinese institutions lead the world in terms of volume of coronavirus articles (including both published articles and preprints) in both the pre- and COVID-19 periods. Moreover, during the COVID-19 period, eight out of top ten of the most prolific institutions are located in China. Wuhan University (which includes Renmin Hospital and Zhongnan Hospital) and Huazhong University of Science and Technology (which includes Tongji Hospital, Tongji Medical College, and Wuhan Union Hospital), located in Wuhan, China are the most prolific institutions during COVID-19, followed by the University of Hong Kong, and Fudan University. However, the Chinese Centers for Disease Control (Chinese CDC), which leads coronavirus research output in the pre-COVID-19 era, drops down the list in the COVID-19 period. In contrast to the rise of some Chinese

**Table 3. Number of publications in COVID-19 vs pre-COVID-19 research, by author country.**

| | Pre-COVID-19 (% of total global articles) January 2018-December 31 2019 | | COVID-19 (% of total global articles) January 1 2020 –April 23 2020 | | COVID-19 (minus preprints) (% of total global articles) January 1 2020 –April 8 2020 | |
|---|---|---|---|---|---|---|
| | Overall | International Teamed Articles | Overall | International Teamed Articles | Overall | International Teamed Articles |
| **Total global articles** | 6,105 | | 4,327 | | 2,472 | |
| **China** | 1,341 (22%) | 469 (7%) | 1,671 (39%) | 507 (12%) | 1,069 (42%) | 332 (13%) |
| **United States** | 2,122 (35%) | 1,129 (18%) | 1,202 (28%) | 533 (12%) | 605 (24%) | 326 (13%) |

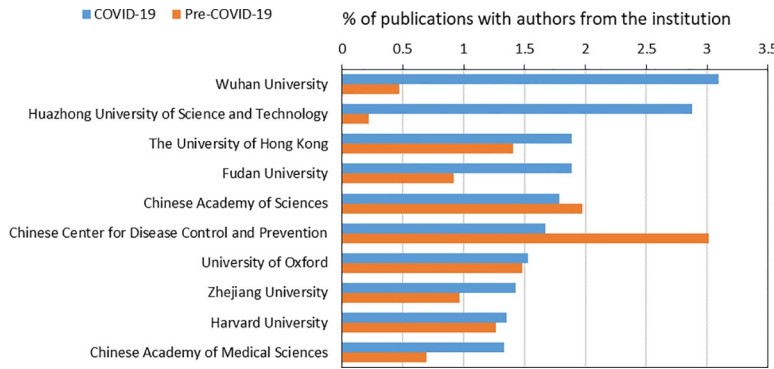

**Fig 1. Top 10 institutions in terms of publication quantity in coronavirus research in the COVID-19 period.**

institutions in the COVID-19 period, the United States National Institutes of Health drops out of the top ten most prolific institutions during COVID-19 at the time of writing of this study. When we exclude preprints from the analysis, we observe that Fudan University drops a lot in ranking and the University of Oxford and Harvard University drop out of the top ten most prolific institutions during the COVID-19 period, leaving the top ten institutions to exclusively consist of Chinese institutions.

## Reported funding for coronavirus research

Next, we examine the most commonly acknowledged funding agencies in coronavirus research before and during the COVID-19 period (Table 4). Self-reported funding data is aggregated where possible across published articles found in the Web of Science (WoS) and Elsevier Scopus database. We find that during the COVID-19 period, Chinese agencies are more likely to be acknowledged as the funding source in published work than before the outbreak. In particular, during the COVID-19 period, the most commonly acknowledged funders are National Natural Science Foundation of China (NSFC) and the Ministry of Science and Technology, China (MOST), which includes the Ministry of Science and Technology (MOST), National Key Research and Development Program of China, National Basic Research Program of China, National High Technology Research and Development Program of China.

**Table 4. Funder of COVID-19 vs pre-COVID-19 research.**

| Funders | Pre-COVID-19 | | COVID-19 | |
|---|---|---|---|---|
| | Rank* | % of funded articles | Rank | % of funded articles |
| **National Institutes of Health (NIH)** | 1 | 16.9% | 3 | 12.2% |
| **National Natural Science Foundation of China (NSFC)** | 2 | 14.3% | 1 | 28.3% |
| **Ministry of Science and Technology, China (MOST)** | 3 | 10.7% | 2 | 12.9% |
| **European Union (EU)** | 4 | 3.3% | N/A | N/A |
| **National Research Foundation of Korea (NRFK)** | 5 | 3.1% | N/A | N/A |
| **Fundamental Research Funds for the Central Universities (FRFCU)** | N/A | N/A | 4 | 3.7% |
| **Japan Society for the Promotion of Science (JSPS) / Ministry of Education, Culture, Sports, Science and Technology (MEXT)** | N/A | N/A | 5 | 3.3% |
| **Chinese Academy of Sciences (CAS)** | N/A | N/A | 5 | 3.3% |

* "N/A" indicates the funder is not in the top 5 list.

**Table 5. Country origins of funder of COVID-19 vs pre-COVID-19 research.**

| Nationality | Pre-COVID-19 | | COVID-19 | |
|---|---|---|---|---|
| | Rank* | % of funded articles | Rank* | % of funded articles |
| **China Mainland** | 1 | 27.0% | 1 | 46.0% |
| **United States** | 2 | 25.8% | 2 | 18.5% |
| **South Korea** | 3 | 6.7% | N/A | N/A |
| **United Kingdom** | 4 | 4.8% | 5 | 3.9% |
| **Europe** | 5 | 4.4% | 3 | 4.4% |
| **Japan** | N/A | N/A | 4 | 4.1% |

* "N/A" indicates the country is not in the top 5 list.

In contrast, in the early days of the COVID-19 pandemic, United States funders are less likely to be cited as the funding agency than before the pandemic. As an example, the United States Department of Health, which includes the National Institutes of Health and its affiliated funding agencies, drops from the most commonly acknowledged funder in coronavirus research before COVID-19 to the third most commonly cited funder during the COVID-19 period.

Table 5 aggregates data on acknowledged funding sources by country-of-origin. During COVID-19 we see Chinese agencies acknowledged as funding the majority of published papers. In this period, at least 46% of articles acknowledge funding from Chinese agencies, while only 18% of publications acknowledge funding from United States based funders. However, prior to the COVID-19 period, Chinese and United States agencies fund about the same number of articles, and this will likely recalibrate as the United States recovers from the initial lockdown. The shift could be due in part to the greater share of Chinese articles during COVID-19, China's longer experience with COVID-19, and support from the Chinese government for coronavirus related research during the COVID-19 pandemic.

## Structure of teams

Our primary research questions are related to the structure of teams and international collaboration following the COVID-19 pandemic. Table 6 reveals the first results pertaining to these research questions. The table shows that during the COVID-19 period, research teams are smaller on average in published articles, although not in preprints. We note that preprint teams are slightly larger, which could be due to the recency of these works, since larger teams

**Table 6. Structure of scientific teams in COVID-19 vs pre-COVID-19 research.**

| | Pre-COVID-19 | | | | COVID-19 | | | | COVID-19 (minus preprints) | | | |
|---|---|---|---|---|---|---|---|---|---|---|---|---|
| | Mean | Median | Std Dev | N | Mean | Median | Std Dev | N | Mean | Median | Std Dev | N |
| **Number of authors** | 7.09*** | 6 | 5.57 | 6,105 | 7.83*** | 6 | 8.58 | 4,327 | 6.38 | 5 | 7.61 | 2,472 |
| **Number of countries** | 1.76** | 1 | 1.34 | 6,106 | 1.52 | 1 | 1.32 | 4,327 | 1.65 | 1 | 1.32 | 2,472 |
| **International team dummy** | 0.42*** | 0 | 0.49 | 6,106 | 0.31 | 0 | 0.46 | 4,327 | 0.35 | 0 | 0.48 | 2,472 |
| **International team dummy (excluding Chinese articles)** | 0.44*** | 0 | 0.50 | 4,764 | 0.32 | 0 | 0.46 | 2,656 | 0.38 | 0 | 0.49 | 1,431 |
| **International team dummy (Chinese articles)** | 0.35** | 0 | 0.48 | 1,341 | 0.30 | 0 | 0.46 | 1,671 | 0.31 | 0 | 0.46 | 1,041 |
| **International team dummy (United States articles)** | 0.53*** | 1 | 0.50 | 2,122 | 0.44 | 0 | 0.50 | 1,202 | 0.54 | 0 | 0.50 | 605 |

*, **, *** denote statistical significance at p values of 0.1, 0.05 and 0.01 in a difference of means test comparing pre-COVID-19 and COVID-19 outcomes. Comparisons are between pre-COVID-19 outcomes and COVID-19 era outcomes.

take longer to produce output. The table also shows that articles in the COVID-19 period are less likely to be internationally coauthored than pre-COVID-19, which is expected given the transaction costs involved in distance collaboration in the early days of the pandemic. The number of countries involved in coauthored articles also drops in the COVID-19 period. This decline in international collaborations is lower for Chinese authors than for the rest of the world, although this difference is not statistically significant. United States-based authors do not experience a change in international collaborations in published articles (minus preprints); however in preprints, we see a decline for United States-based researchers at the international level.

## Journal impact of peer-reviewed research

One concern that may arise from this rapid explosion of articles and changing team structures in coronavirus research during the global pandemic is a very broad range of quality than prior to the crisis. A review of impact factors attached to journals carrying coronavirus publications shows that these works are actually published in higher impact journals than was the case in the pre-COVID-19 period. To test for impact, we weight each published article in the two data-sets by the Elsevier Scopus Source Normalized Impact per Paper (SNIP) of the publication journal, which is calculated as the journal's citation count per paper divided by its citation potential in its subject area. We assess whether articles in the COVID-19 period are, on average, published in higher impact journals than pre-COVID-19 articles, and whether the growth in Chinese authored publications is driven by publication in lower impact journals. Results are shown in Table 7.

The positive coefficient on COVID-19 in Table 7 column 1 reveals that articles in our sample are published in journals with higher SNIP values in the COVID-19 period compared to the pre-COVID-19 period. This suggests that journal editors and peer reviewers acted quickly in response to the need for scientific understanding about the novel coronavirus. Chinese-authored publications are appearing in as high impact journals as the rest of the world in both the pre- and COVID-19 periods (column 2), and publications with international teams appear in significantly higher impact journals than those with domestic-only teams (column 3). Column 4 reveals that Chinese authors publish in higher-impact journals during the COVID-19 era than the rest of the world although the difference is not statistically significant.

**Table 7. Regression analysis of the relationship between team structure and the impact factor of journals publishing coronavirus research in pre- and during COVID-19.**

| Independent variables | Dependent variable—Source Normalized Impact per Paper | | | | |
|---|---|---|---|---|---|
| | (1) | (2) | (3) | (4) | (5) |
| COVID-19 | 0.086*** (0.020) p = .000 | 0.088*** (0.019) p = .000 | 0.088*** (0.019) p = .000 | 0.064*** (0.024) p = .007 | 0.107*** (0.027) p = .000 |
| Authors China | | -0.009 (0.013) p = .459 | 0.0012 (0.013) p = .921 | -0.023** (0.012) p = .044 | |
| International Team | | | 0.069*** (0.012) p = .000 | | 0.078*** (0.011) p = .000 |
| COVID-19 x Authors China | | | | 0.062 (0.041) p = .127 | |
| COVID-19 x International Team | | | | | -0.046 (0.039) p = .235 |
| N | 4,502 | 4,502 | 4,502 | 4,502 | 4,502 |

Estimates stem from ordinary least square model regression specifications with dependent variables being inverse hyperbolic sine transformed SNIP of a publication in the sample, and independent variables being the period of the publication (COVID-19 or pre-COVID-19) (column 1), whether the authors of the publication are from a Chinese institution (column 2), and whether the publication author team is international (column 3). In columns 4, 5, and 6 we include interaction terms of COVID-19 period and the team structure to assess whether there is a different relationship between team structure and SNIP of a publication pre and during-COVID-19.

Robust standard errors in parentheses.

*, **, *** denote statistical significance at p values of 0.1, 0.05 and 0.01.

Table 7, column 5 shows that although internationally coauthored articles are published in higher impact journals than domestically teamed ones, there is no differential increase in the COVID-19 era for international versus domestic teams. Together with the documented decline in international collaborations, this result suggests that domestic teams during COVID-19 are increasing the impact factor of the journals they publish in as compared to before.

### International networks of collaboration

Network collaborative patterns have shifted in the COVID-19 era, as expected. Table 8 shows network metrics for major actors in the global network in pre-COVID-19 and COVID-19. The United States is the core player in both the pre-COVID-19 and COVID-19 networks. However, with a decreasing measure of collaboration linkages, or degree, during the COVID-19 era, the United States shows a decreasing role in the network. This is particularly true in comparison to China and Italy, which show increased collaboration linkages during the COVID-19 period, as measured by degree. Among the nations listed, China, the United Kingdom, and Italy show an increase in their normalized betweenness centrality during the COVID-19 pandemic, as compared to the 24-months leading up to the pandemic, indicating an increase in their bridging roles in the network. In terms of eigenvector centrality, the top four nations—United States, China, the United Kingdom, and Italy—retain their centrality into the COVID-19 period but Germany drops considerably in the early days of COVID-19, perhaps due to a lag in research output.

Figs 2 and 3 reveal the network structures in coronavirus research among nations in pre-COVID-19 and COVID-19 periods. Fig 2 illustrates that the United States is at the core of the international collaborative network in coronavirus research before the COVID-19 pandemic, while China is the third most central country in the network. Given the dominance of the United States in terms of research output in the pre-COVID-19 period, this is not a surprise, as the networks are not normalized by size. The figure also shows that connections between the United States and China before the crisis constitute the strongest bilateral link in the field, while there are also strong connections between the United States and the United Kingdom, France, Germany, Saudi Arabia, India, and Canada.

Fig 3 presents the network of international collaboration in the COVID-19 period. As expected, given the shorter period available for production of research, the network is sparse compared to the pre-COVID-19 period. However, we can see that the United States-China relationship has intensified compared to the pre-COVID-19 period. Moreover, these two nations maintain their status as the most centralized players in the collaborative network. Scientifically advanced nations like the United Kingdom, Germany, France, Italy, and Canada remain active in the network, although their strongest connections are still with the United States. Germany's role in the network declines considerably, as do many other nations.

**Table 8. Network metrics for selected nations in COVID-19 and pre-COVID-19.**

| | Pre-COVID-19 (network based on 2,072 articles) | | | | COVID-19 (network based on 1,112 articles) | | | |
|---|---|---|---|---|---|---|---|---|
| | Degree | Weighted Degree | Normalized Betweenness Centrality | Eigenvector centrality | Degree | Weighted Degree | Normalized Betweenness Centrality | Eigenvector centrality |
| **United States** | 97 | 1,915 | 0.163 | 1 | 74 | 907 | 0.204 | 1 |
| **China** | 62 | 687 | 0.023 | 0.802 | 52 | 698 | 0.063 | 0.840 |
| **United Kingdom** | 82 | 992 | 0.064 | 0.954 | 66 | 523 | 0.129 | 0.952 |
| **Italy** | 61 | 364 | 0.015 | 0.805 | 49 | 271 | 0.059 | 0.806 |
| **Germany** | 78 | 639 | 0.050 | 0.930 | 55 | 261 | 0.067 | 0.873 |

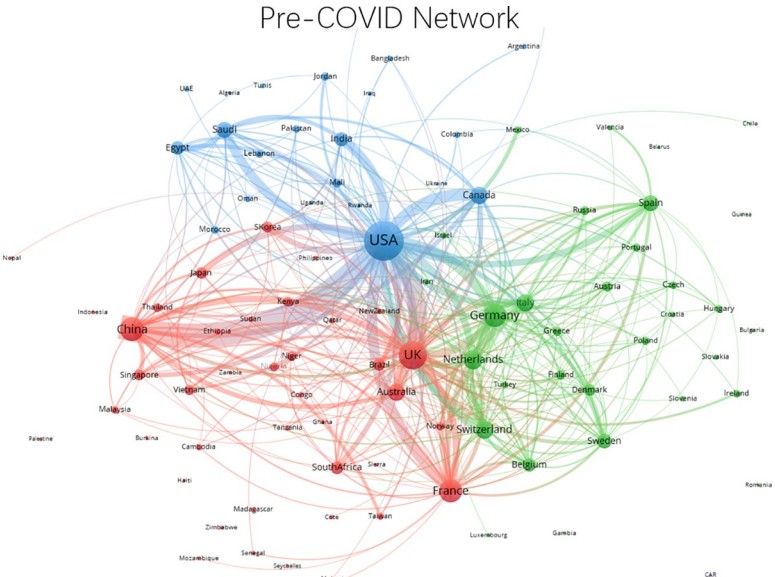

**Fig 2. Network of international collaborative relationships in pre-COVID-19 period, January 1, 2018 to December 31, 2019.** Edges lower than 4 are removed.

Fig 4A and 4B present the egonets with China as the central player to illustrate China's collaborative pattern in the pre- and COVID-19 periods. The figures further support the observation that China's strongest collaborative link is with the United States and that this relationship intensifies in the COVID-19 period. Overall, the share of all coronavirus publications that comprise China-United States collaborations increases to 4.9% from 3.6% in the

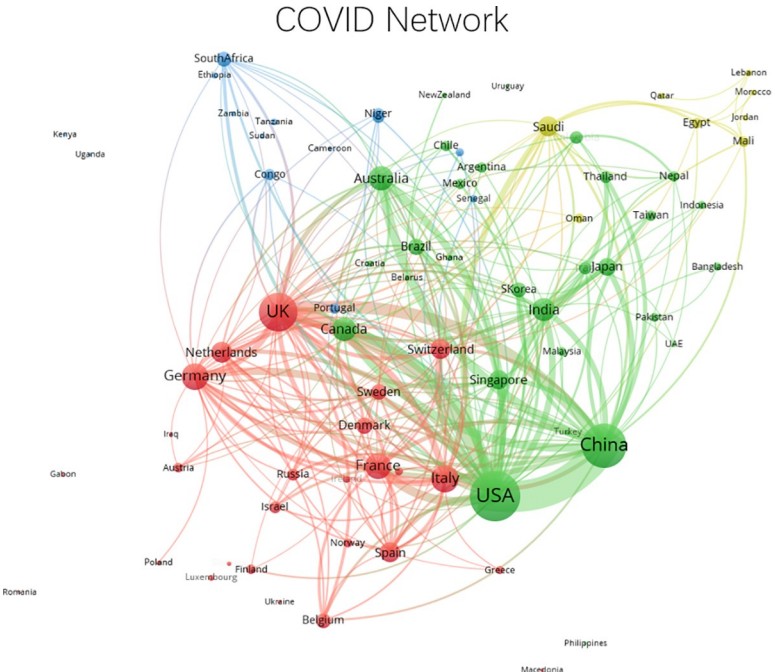

**Fig 3. Network of international collaborative relationships during COVID-19 period, January 1, 2020 to April 8, 2020.** Edges lower than 2 are removed.

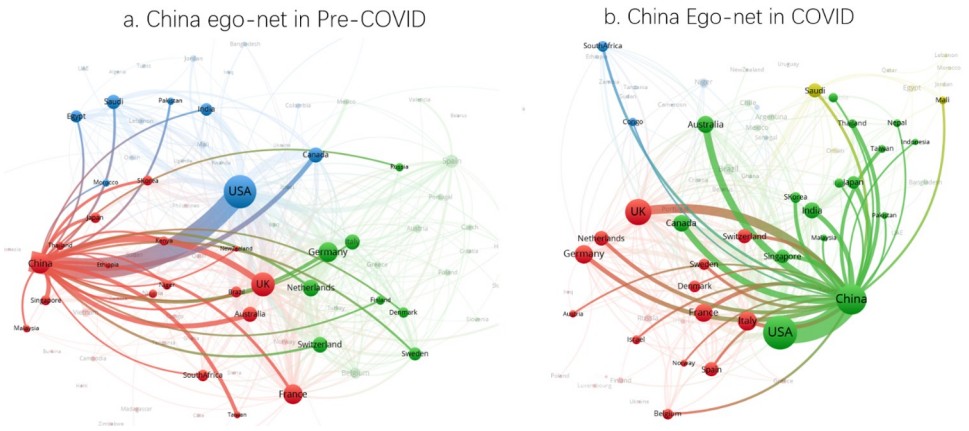

**Fig 4. Ego-networks of China in pre-COVID-19 and COVID-19 periods.**

COVID-19 period. Table 9 columns 1 and 2 confirms the increase in the rate of China—United States collaborations as statistically significant in the COVID-19 period in all coronavirus articles and all internationally collaborative coronavirus articles.

That said, because of China's increased quantity of publications in the COVID-19 period, the relative share of collaboration with the United States as a function of overall Chinese publications drops, as shown in Table 9 columns 3 and 4. In an absolute sense, China is producing more and higher quality work on its own. In addition to increasing their domestic outputs, China also strengthened links with Canada, Japan, the Netherlands, Italy, and India in an effort to advance COVID-19 research worldwide.

Similarly, Fig 5A and 5B show the egonet diagram of the United States as the central player, and further support the observation that during COVID-19, the United States has solidified its relationships with a handful of specific countries, particularly China. The consolidation of the United States—China relationship is closely related to dominant role of China in articles published during the COVID-19 pandemic. The United States' collaboration with China remains its strongest link; this can also be seen in Table 9 columns 5 and 6. In contrast, the United States' relative share of collaborative articles with many other nations, such as the United

**Table 9. Regression analysis of the rate of pairwise collaboration between China and the United States in COVID-19 and pre-COVID-19 research.**

| Independent variable | Dependent variable—China-United States Collaboration | | | | | |
|---|---|---|---|---|---|---|
| | **(1)** | **(2)** | **(3)** | **(4)** | **(5)** | **(6)** |
| COVID-19 | 0.014*** (0.004) | 0.069*** (0.012) | -0.060*** (0.014) | -0.055 (0.036) | 0.091*** (0.014) | 0.20*** (0.026) |
| Sample | Full | Internationally collaborative articles | Chinese authored articles | Chinese authored articles with international collaborations | United States authored articles | United States authored articles with international collaborations |
| **p-value** | .001 | .000 | .000 | .131 | .000 | .000 |
| **N** | 10,432 | 3,915 | 3,012 | 976 | 3,324 | 1,662 |

Estimates stem from linear probability models specifications with dependent variables being dummy variables taking the value of 1 if the article contains a China-United States collaboration and independent variable being COVID-19 period or pre-COVID-19. All models include controls for type of article (formal/informal). The samples for the regression in model 1 is the full sample of articles, model 2 is just the set of internationally collaborative articles, model 3 is just Chinese authored articles, 4 is just Chinese articles authored with international collaborators, model 5 if just United States authored articles and model 6 is just United States authored articles with international collaborators.

Robust standard errors in parentheses.

*, **, *** denote statistical significance at p values of 0.1, 0.05 and 0.01.

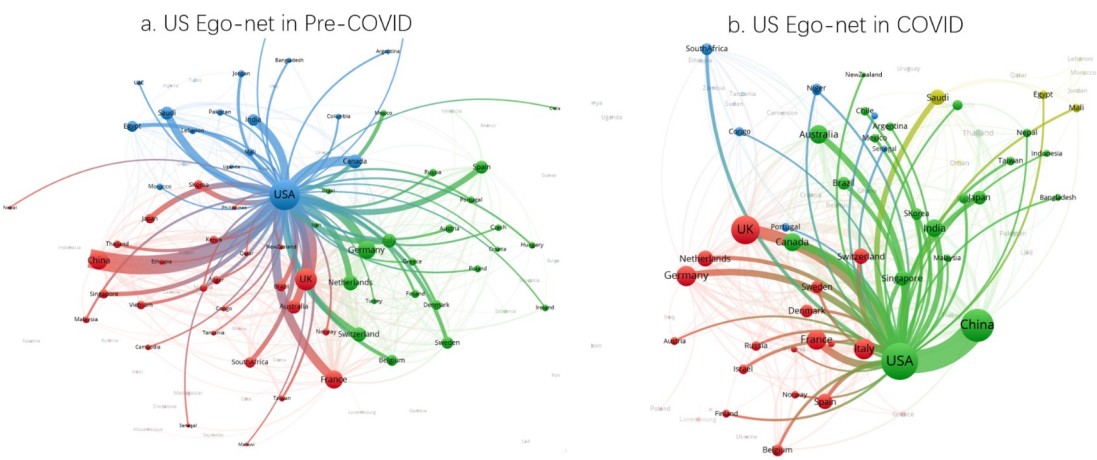

**Fig 5. Ego-networks of the United States in COVID-19 and pre-COVID-19 periods.**

Kingdom, India, Canada, Germany, France, and Australia, and developing countries declines considerable during COVID-19.

## Research topics in coronavirus articles

Research topics identified from coronavirus articles in pre-COVID-19 and COVID-19 periods provide clues to the potential changes of research emphases during the crisis. Fig 6A reveals

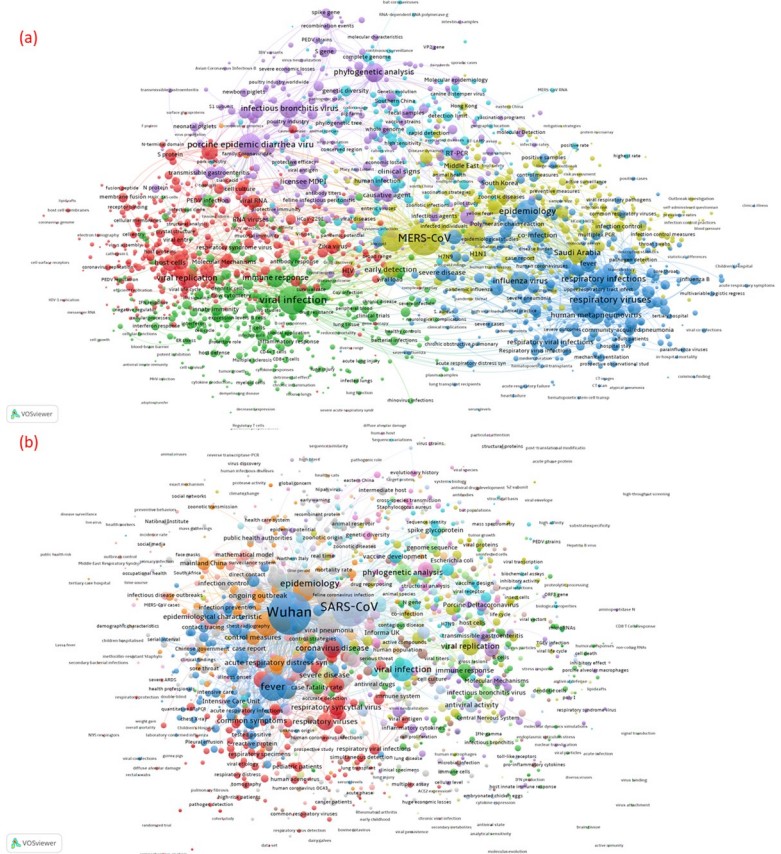

**Fig 6. Co-term networks of the COVID-19 research in pre- and COVID-19 periods.**

five main clusters of research topics that appear in coronavirus articles in the pre-COVID-19 period, including viral replication (red nodes), viral infection (green nodes), respiratory infections (blue nodes), public health topics (yellow nodes and epidemiology-related blue nodes), and molecular epidemiology (purple nodes). We interpret the relatively clear boundaries between these clusters as evidence for the existence of established research area with well-defined concepts and explicit topics. Comparably, Fig 6B illustrates the focus on a more diverse, and 'chaotic' set of research topics during the early COVID-19 period. The largest node representing a research topic is "Wuhan", reflecting the location of the first occurrence of the virus. The network reveals a diverse range of topic pursued by researchers during COVID-19 including epidemiological characteristics, symptom descriptions, geographical features, and public health concerns (blue and orange nodes). This indicates the predominant concerns amongst researchers, but also reveals that researchers lack a clear focus and coordination. We expect this to change as the features of the virus, the public health concerns, and patient care practices become clearer.

## Discussion

Science is increasingly a team activity [15], with scientists self-organizing into collaborations, including international collaborations, as needed by the research questions [6,16]. In particular, the involvement of many more countries coming into the global network of scientists over the past 20 years has led to dramatic shifts in the structure of scientific activity around the world [7]. On the one hand, international collaboration can allow scientists to access expertise, funding, and resources outside of their own nation. However, the search and coordination costs of this type of collaboration are high. For scientists, the decision to engage in international collaboration represents this inherent trade-off.

During a global pandemic these trade-offs intensify. The need for broad expertise and pooled resources is greater than ever, but, an urgent need for scientific input into public health and economic decisions puts a premium on transaction costs associated with long-distance and cross-cultural communications. Based on this logic, in this study we hypothesize that a global pandemic would result in a reduction in the usual search and outreach, since scientists need to limit the coordination costs of research. Specifically, we expect to see that researchers return to known-collaborators and smaller teams to speed the research process during a global pandemic. We test these hypotheses through comparing the patterns found in coronavirus related research prior to and during the COVID-19 global pandemic and find that the pandemic is inducing changes in the global organization of science, at least as related to coronavirus.

A review of early publication and cooperation patterns of scientific publications at the global level highlights that scientists rapidly reorganized to address the crisis posed by COVID-19 along the lines of greater efficiency and narrower focus. As expected, the dynamics of collaboration and teaming appear to shift to rely on fewer team members, which reduces the transaction costs of communicating among the group, and can, in theory, speed the research and writing processes. The challenge of the novel virus strengthens the research relationship between China and other scientifically advanced countries, especially with the United States. At the same time, Chinese researchers become more independent increasing the volume, and quality, of domestic output in the COVID-19 era. Moreover, the Chinese research funding agencies played a vital role in the earliest days in supporting high quality research and development work in China. These findings are in contrast to some popular accounts that Chinese scientists are withholding valuable information and reducing cooperation in the early stages of the global pandemic [17].

Although we interpret the findings as providing insight into the theoretical expectations, this study has three main limitations. First, the period used for analysis pre-COVID-19 and during COVID-19 is different. Due to the nature of data collection, we had a much shorter period available to us for the COVID-19 period. We account for this in the statistical analysis, however we anticipate that some of our trends may change longer term. In particular, the geographic nature of scientific production and funding may represent the geographic spread of the coronavirus. For example, the virus spread through China before it was observed in the United States, and so it is not a surprise that the Chinese government and funding agencies allocated resources earlier than the United States. Future research should explore longer term dynamics in scientific funding and productivity around the world.

Second, we use the Source Normalized Impact per Paper (SNIP) to reflect the impact of research. There are many possible measures of research impact, including raw citations, normalized citation scores, and practical measures such as policy influence, news take-up and discussions in online forums. Despite the limitations with the use of SNIP to assess impact, including the influence of publication language, open access journals, fast track publications and the fact that publications from zero impact journals are excluded [18], we chose this measure due to the real time data collection precluding the use of citation measures and alternative measures of influence. Future research should use a variety of impact measures to identify the most important and impactful research during the COVID-19 period.

Finally, our study is limited by the data available on scientific production and funding. In particular, our measure of funding source for coronavirus research is limited. We exploit the text in funding acknowledgments in articles, many of which do not have funding acknowledgments as it is not a requirement by the journal or database. Future work could use data directly from funders themselves as it becomes available or surveys of researchers to ascertain any trends.

That said, we interpret the findings from this study as providing insight into the theoretical framework, and this paper suggests that global scientists face a trade-off in decisions on international collaborative activities around time and efficiency, and that the trade-off changes dramatically during a time of urgency, such as the COVID-19 pandemic. The observed reduction in the rate of international collaborations and consolidation of the strongest existing bilateral relationships during the COVID-19 pandemic could have consequences for the organization of science and direction of research. One of the most important of these findings is a reduction in participation of researchers from developing countries in coronavirus related research. Future work will examine the nature of teaming, preferential attachment, the role of influential individuals in the global networks as well as the consequences of the organization of science on the evolution of research topics during the COVID-19 pandemic.

Policymakers who are tracking and guiding research into coronavirus topics and vaccines may wish to be aware of the changing dynamics of international teams. While it is important to increase efficiency, smaller teams could mean that knowledge diffusion and wide-ranging expertise and novelty are reduced [3,15]. This fact is particularly true for those research institutions which are not among the most elite institutions or within those teams gathered around leading scientists who have ample funding. The results of the narrowing and focusing of research may mean that results arrive more quickly, but it also means the results and capacities are diffused more slowly. Validation may be compromised. Policy actions to address these inequities may be needed in the very near term.

## Author Contributions

**Conceptualization:** Yi Zhang, Caroline S. Wagner.

**Data curation:** Yi Zhang.

**Formal analysis:** Caroline V. Fry, Xiaojing Cai, Caroline S. Wagner.

**Methodology:** Caroline V. Fry.

**Software:** Yi Zhang.

**Supervision:** Caroline S. Wagner.

**Validation:** Yi Zhang.

**Visualization:** Xiaojing Cai.

**Writing – original draft:** Caroline V. Fry, Caroline S. Wagner.

**Writing – review & editing:** Caroline S. Wagner.

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
