## [Decision Letter · Decision Letter 0]

10 Jun 2020

PONE-D-20-13949

Consolidation in a Crisis: Patterns of International Collaboration in COVID-19 Research

PLOS ONE

Dear Dr. Wagner,

Thank you for submitting your manuscript to PLOS ONE. After careful consideration, we feel that it has merit but does not fully meet PLOS ONE’s publication criteria as it currently stands. Therefore, we invite you to submit a revised version of the manuscript that addresses the points raised during the review process. Please submit your revised manuscript by Jul 25 2020 11:59PM. If you will need more time than this to complete your revisions, please reply to this message or contact the journal office at plosone@plos.org. Please include the following items when submitting your revised manuscript:

We look forward to receiving your revised manuscript.

Kind regards,

Lutz Bornmann

Academic Editor

PLOS ONE

Journal Requirements:

Reviewers' comments:

Reviewer's Responses to Questions

**Comments to the Author**

1. Is the manuscript technically sound, and do the data support the conclusions?

Reviewer #1: Yes

Reviewer #2: Yes

2. Has the statistical analysis been performed appropriately and rigorously? 

Reviewer #1: Yes

Reviewer #2: Yes

3. Have the authors made all data underlying the findings in their manuscript fully available?

Reviewer #1: Yes

Reviewer #2: No

4. Is the manuscript presented in an intelligible fashion and written in standard English?

Reviewer #1: Yes

Reviewer #2: Yes

5. Review Comments to the Author

Reviewer #1: This is a well written manuscript which addresses several dimensions of collaboration and also has important policy implications.

Abstract - I do feel that this type of study would benefit from a structured abstract with more description of quantative results. It would be easier to understand the concept of the study.

Introduction - Although the introduction is well written, it is too long and which makes it difficult to understand what the study aims are. Perhaps the authors could make this section more concise clearly stating the aims.

Methods

Publication patterns and numbers

Table 2 and table 3 correctly describe results with preprints excluded but table 4 does not. I think it is important to distinguish between peer reviewed publications and preprints. Moreover table 4 is confusing because it is difficult to interpret the rankings pre covid-19 vs covid-19. It maybe better to present in graphical form. If preprints were excluded do the rankings change?? Perhaps lower quality research is over represented in the covid-19 category compared to pre covid-19?

Public funding patterns to compare pre- and COVID-19

The main reason for differences in funding are probably related to the time point of the pandemic the country was experiencing. For example the pandemic peaked in the USA whilst China was in the recovery phase. Chinese government/funding agencies would have allocated resources much earlier than the USA. If the study is repeated now I am sure that funding for covid-19 research in the USA will have increased exponentially. The authors need to clarify this.

Quality measures of formal publications

Using journal impact factor as a measure of the quality of publication has its flaws. Please see: How has healthcare research performance been assessed? A systematic review - Patel et al Journal of the Royal Society of Medicine; 2011 104(6): 251-261. Authors need to state this as a limitation.

Collaborative patterns at the international level and networked collaborations including measures of egonets

at the international level.

The network perspective of this study is the most interesting. The authors have used degree and betweenness to measure collaboration. What are the reasons for choosing these measures? There are several other network measures that can be used such as eigenvector and closeness as well as social capital measured with clustering coefficient. Please see:Collaborative patterns, authorship practices and scientific success in biomedical research: a network analysis Patel et al. 2019 112(6): 245-257.Journal of the Royal Society of Medicine; 104(6): 251-261. The authors should explain the rationale for choosing the network metrics they have used in the analysis and state the limitations.

Discussion

This type of study has its strengths and limitations and this should be included in the discussion. Any limitations outlined in the results should be moved to the discussion to avoid repetition.

Reviewer #2: The manuscript "Consolidation in a crisis: patterns of international collaboration in COVID-19 research" presents an analysis of international collaboration patterns for COVID-19 papers in comparison to other coronavirus-related research in pre-COVID-19 times. The manuscript is rather well-written (although it would benefit from proof-reading, see below) and should be of interest to the readers of PLoS One. However, some improvements should be made before publication of the manuscript.

The distinction between "pre-COVID-19" and other papers is inconsistent. The other papers are sometimes referred to as "COVID-19", "during COVID-19", and "post-COVID-19". The latter is problematic because we are not yet in a "post-COVID-19" time. I recommend to remove the "post-" and settle with "during COVID-19". I guess "during COVID-19" should be hyphenated when "pre-COVID-19" is. There are also references to "pre-COVID" and "pre-COVID-19". I recommend unifying the usage to "pre-COVID-19".

"The pre-COVID-19 period extends for 24 months prior to December 2019. The COVID-19 period extends from January 1, 2020 to April 23, 2020." (lines 111-113) These two data sets cover rather differently sized time frames: 24 month vs. four months. One time frame is six times larger than the other. Four months is a very short time window for publication output analysis. Are the results still comparable although the paper sets are of a similar size? Both paper sets span different time scales and cover somewhat different topics. This should be discussed as one of the limitations of the study.

Maybe networks of keywords or title words can help to find out if the topics of both studied time periods are similar or different.

Collection of the data set is described in lines 116 ff. I wonder if additional "during-COVID-19" papers could have been found in the WHO Global research database on coronavirus disease (COVID-19): https://www.who.int/emergencies/diseases/novel-coronavirus-2019/global-research-on-novel-coronavirus-2019-ncov

Is there a reference to cite the quote from Collins in lines 95-98?

It is unclear to me how the percentages in Table 3 are calculated, e.g., the last column has a higher absolute number for China than for USA whereas it shows a lower percentage for China than for USA. The table heading reads "(%of total global articles)" in all columns. If 332 papers are 37%, how can 1069 papers be 42%?

The section "Impact of peer-reviewed research" should be reworded more carefully. In the current version, journal impact is equated with publication impact.

I do not understand Table 8. Please provide a better explanation. The caption states "Publication Impact in COVID-19 vs pre-COVID-19 Research". The header states "Source Normalized Impact per Paper" but I do not see a SNIP. The footer and data in the table suggest some regression analysis. What does the "X" in "COVID-19 X Authors China" mean?

Similarly, Table 10 should be explained better. The caption reads "Pairwise Collaboration Rates of Nations in COVID-19 and Pre-COVID-19 368 Research" although the values in the table and its footer indicate regression results. The header of Table 10 ("Pairwise Rate of China/USA Collaborations") indicates that only papers from China and USA are analyzed whereas the caption indicates a broader analysis including more countries.

The authors used VOSviewer for visualizations. The map and net files could be shared with readers.

Finally, the manuscript would benefit from proof-reading:

- "bioXriv.org, medrX" ext-link-type="uri" xlink:type="simple">iv.org" -- "bioRxiv.org, medRx" ext-link-type="uri" xlink:type="simple">iv.org" (line 121)

- "China also strengthened links with the Canada, Japan, Netherlands, Italy, and India ..." -- "China also strengthened links with Canada, Japan, the Netherlands, Italy, and India ..." (lines 381/382)

- "... the pandemic is induces changes ..." -- "... the pandemic is inducing changes ..." (line 417)

6. PLOS authors have the option to publish the peer review history of their article (what does this mean?). If published, this will include your full peer review and any attached files.

Reviewer #1: No

Reviewer #2: No

---

## [Author Response · Author response to Decision Letter 0]

19 Jun 2020

Lutz Bornmann

Academic Editor

PLOS ONE

Dear Lutz, 

Thank you for the consideration of our paper “Consolidation in a Crisis: patterns of international collaboration in early COVID-19 research”. On behalf of my coauthors, Yi Zhang, Xaiojing Cai, Caroline Fry and myself, we appreciate the comments from the reviewers and their time in reading and reviewing this paper. We have carefully considered the comments and responded to them as fully as possible in the attached manuscript, with detailed responses to each comment outlines in this paper. We are happy to resubmit the revised paper, with text below that addresses the requested changes. 

We thank reviewer #1 very much for their careful reading of the manuscript, and insightful comments. First, reviewer #1 suggested some editorial changes – such as a ‘structured abstract with more description of quantitative results’ and a suggestion to shorten the introduction clearly stating aims of the study. We have adapted the abstract and shortened the introduction, tightening the language in both, clearly highlighting the goals of the study. 

Reviewer #1 notes that “The main reason for differences in funding are probably related to the time point of the pandemic the country was experiencing. For example the pandemic peaked in the USA whilst China was in the recovery phase. Chinese government/funding agencies would have allocated resources much earlier than the USA. If the study is repeated now I am sure that funding for covid-19 research in the USA will have increased exponentially.” We really appreciate this comment, and have clarified this important feature of our data in the discussion section of the revised paper. 

Reviewer #1 also suggests that we state the limitation of using journal impact factors as a measure of quality. We do so in a limitations section in the discussion, which we add on the suggestion of the reviewer, and we greatly appreciate the citation the reviewer suggested to a paper outlining the limitations of using quality measures and have included this reference in our paper. 

Reviewer #1 notes that “Table 2 and table 3 correctly describe results with preprints excluded but table 4 does not. I think it is important to distinguish between peer reviewed publications and preprints. Moreover table 4 is confusing because it is difficult to interpret the rankings pre covid-19 vs covid-19. It maybe better to present in graphical form.” We appreciate this comment, and have actually substituted the table for a graphic (figure 1) which we believe illustrates the changing dominance of institutions in coronavirus research pre and during COVID-19 in a more effective manner. In response to the reviewer’s suggestion, we also clarify the language in our analysis that includes preprints and the analysis that excludes them, and any differences between the results. This was a great suggestion from the reviewer, as we discovered without preprints, the top 10 producing institutions in the world in COVID-19 were all Chinese institutions. 

Reviewer #1 asks the rational for choosing network measures degree and betweenness and suggested a helpful paper outlining some other potential network measures we could consider. We thank the reviewer for this comment, and added an additional measure of eigenvector centrality to the analysis, as well as included a summary of our rationale on the use of centrality, adding an additional citation in addition to the reference that the reviewer suggested. 

We thank reviewer #2 for their careful reading of the paper. They suggested some proof reading, particularly to ensure term consistency (COVID-19, during COVID-19), and to include a reference to a quotation we use in the introduction. We thank the reviewer very much for this comment, and their suggestion on the most useful terms to use throughout, and have carefully proof read and edited the manuscript, including the reference to the quotation used. 

Reviewer #2 noted, "The pre-COVID-19 period extends for 24 months prior to December 2019. The COVID-19 period extends from January 1, 2020 to April 23, 2020." (lines 111-113) These two data sets cover rather differently sized time frames: 24 month vs. four months. One time frame is six times larger than the other. Four months is a very short time window for publication output analysis. Are the results still comparable although the paper sets are of a similar size?” We are very grateful for this articulation of one of the features of our data collection strategy from reviewer #2, and on their suggestion we include a discussion of this limitation in the discussion section. 

Reviewer #2 suggests “Maybe networks of keywords or title words can help to find out if the topics of both studied time periods are similar or different.” We thank the reviewer very much for this fantastic suggestion. We add this into the manuscript as an additional figure with associated analysis on the topics pursued in coronavirus research before and during the COVID-19 pandemic, and think that it strengthens the study enormously. 

Reviewer #2 suggested looking in the WHO Global research database for additional publications or preprints during COVID-19. We thank the reviewer for this suggestion, and we feel confident that the data we have captures some of the WHO data, although the affiliation data of many WHO publications are not accessible and so these publications are unusable in much of our analysis, and that our resulting datasets incorporate a complete set of publications during COVID-19. 

Reviewer #2 notes “It is unclear to me how the percentages in Table 3 are calculated”. We apologize for this lack of clarity in how we present the results, and appreciate the reviewer pointing it out to us. We were originally presenting the percentage of all articles, and of just international teamed articles, but in response to the reviewers comment we changed table 3 to present just percentages of all coronavirus research in the time period specified. We think that this allows for much easier interpretation, and still conveys the same message as before. 

Reviewer #2 notes that “The section "Impact of peer-reviewed research" should be reworded more carefully. In the current version, journal impact is equated with publication impact.” We thank the reviewer for this comment, and changed the wording in the text accordingly so as to be precise that we are presenting the journal impact quality and not the impact of the article itself. Following the suggestion of both reviewers on this topic, we also discuss the limitations of the use of the journal impact measures in the discussion section. 

We thank reviewer #2 for their comments on the ease of interpretation of both tables 8 and 10. We made changes to the captions for both of the tables, expanded the table notes and the associated text. We believe that these tables are now much clearer to understand and thank the reviewer for this comment. 

We trust that the revisions made to the paper address the comments of both reviewers, and feel very confident that the comments have allowed us to improve the paper. Thank you again for giving us the opportunity to review and revise this manuscript. 

With kind regards, 

Caroline Wagner

---

## [Decision Letter · Decision Letter 1]

7 Jul 2020

Consolidation in a Crisis: Patterns of International Collaboration in COVID-19 Research

PONE-D-20-13949R1

Dear Dr. Wagner,

We’re pleased to inform you that your manuscript has been judged scientifically suitable for publication and will be formally accepted for publication once it meets all outstanding technical requirements.

Kind regards,

Lutz Bornmann

Academic Editor

PLOS ONE

Additional Editor Comments (optional):

Reviewers' comments:

Reviewer's Responses to Questions

**Comments to the Author**

1. If the authors have adequately addressed your comments raised in a previous round of review and you feel that this manuscript is now acceptable for publication, you may indicate that here to bypass the “Comments to the Author” section, enter your conflict of interest statement in the “Confidential to Editor” section, and submit your "Accept" recommendation.

Reviewer #1: All comments have been addressed

Reviewer #2: All comments have been addressed

2. Is the manuscript technically sound, and do the data support the conclusions?

Reviewer #1: Yes

Reviewer #2: Yes

3. Has the statistical analysis been performed appropriately and rigorously? 

Reviewer #1: Yes

Reviewer #2: Yes

4. Have the authors made all data underlying the findings in their manuscript fully available?

Reviewer #1: Yes

Reviewer #2: No

5. Is the manuscript presented in an intelligible fashion and written in standard English?

Reviewer #1: Yes

Reviewer #2: Yes

6. Review Comments to the Author

Reviewer #1: I am grateful for the authors revising the paper and addressing my comments. The manuscript reads well and is much improved.

Reviewer #2: (No Response)

7. PLOS authors have the option to publish the peer review history of their article (what does this mean?). If published, this will include your full peer review and any attached files.

Reviewer #1: **Yes: **Vanash Patel

Reviewer #2: No

---

## [Editor Report · Acceptance letter]

13 Jul 2020

PONE-D-20-13949R1 

Consolidation in a Crisis: Patterns of International Collaboration in COVID-19 Research 

Dear Dr. Wagner:

I'm pleased to inform you that your manuscript has been deemed suitable for publication in PLOS ONE. Congratulations! Your manuscript is now with our production department. 

Kind regards, 

on behalf of

Dr. Lutz Bornmann 

Academic Editor

PLOS ONE